# The Impact of Selenium Supplementation on Trauma Patients—Systematic Review and Meta-Analysis

**DOI:** 10.3390/nu14020342

**Published:** 2022-01-14

**Authors:** Jen-Fu Huang, Chih-Po Hsu, Chun-Hsiang Ouyang, Chi-Tung Cheng, Chia-Cheng Wang, Chien-Hung Liao, Yu-Tung Wu, Chi-Hsun Hsieh

**Affiliations:** Division of Trauma and Emergency Surgery, Chang Gung Memorial Hospital, Chang Gung University, Taoyuan 33328, Taiwan; jenfuhaung@gmail.com (J.-F.H.); chihpo1227@gmail.com (C.-P.H.); detv090@gmail.com (C.-H.O.); atong89130@gmail.com (C.-T.C.); m0827@cgmh.org.tw (C.-C.W.); hsieh0818@cgmh.org.tw (C.-H.H.)

**Keywords:** trace element, selenium, trauma, injury

## Abstract

This study aimed to assess current evidence regarding the effect of selenium (Se) supplementation on the prognosis in patients sustaining trauma. MEDLINE, Embase, and Web of Science databases were searched with the following terms: “trace element”, “selenium”, “copper”, “zinc”, “injury”, and “trauma”. Seven studies were included in the meta-analysis. The pooled results showed that Se supplementation was associated with a lower mortality rate (OR 0.733, 95% CI: 0.586, 0.918, *p* = 0.007; heterogeneity, *I*^2^ = 0%). Regarding the incidence of infectious complications, there was no statistically significant benefit after analyzing the four studies (OR 0.942, 95% CI: 0.695, 1.277, *p* = 0.702; heterogeneity, *I*^2^ = 14.343%). The patients with Se supplementation had a reduced ICU length of stay (standard difference in means (SMD): −0.324, 95% CI: −0.382, −0.265, *p* < 0.001; heterogeneity, *I*^2^ = 0%) and lesser hospital length of stay (SMD: −0.243, 95% CI: −0.474, −0.012, *p* < 0.001; heterogeneity, *I*^2^ = 45.496%). Se supplementation after trauma confers positive effects in decreasing the mortality and length of ICU and hospital stay.

## 1. Introduction

Traumatic injury accounts for the major causes of death worldwide, and the trimodal of trauma mortality is well established [1]. The first and second peaks were difficult to overcome due to the trauma event itself [2]. However, the third peak in the trimodal model, which is late deaths that occur days to weeks after injury, has become less prominent with the progress of medicine [3]. In this phase, multiorgan failure resulting from infection and sepsis has become one of the major causes of mortality [4,5,6]. Moreover, patient condition is impacted by an initial cascade of inflammation and then aggravated by sepsis [5]. Importantly, advancements in sepsis treatment and critical care strategies have led to the reduction in late mortality [7].

Significant inflammatory responses and severe metabolic disturbances have been noted after major trauma [8]. Selenium (Se) is a key player in regulating immunity and inflammation responses and an essential micronutrient required for more than 25 proteins in the body [9]. These proteins have many functions, including antioxidant defense, protein folding [9], thyroid hormone metabolism [10], and immune health. Moreover, Se has proven to be beneficial in the prevention and recovery of many diseases, such as autoimmune thyroiditis, cancer, and heart disease [11,12]. Reduced Se intake is associated with poor health across the lifespan, increasing Se requirements during pregnancy and lactation and in older individuals [13]. A lack of trace elements could result in sepsis, delayed wound healing, and muscle catabolism. The role of trace elements in major trauma and subsequent sepsis remains unclear.

A lower Se level was found rapidly after trauma events [14,15]. Se plays a critical physiological role in immune function, wound healing, and protein folding and elicits antioxidative effects, which are rapidly depleted after trauma insults [16,17]. Se deficiencies are assumed to result from hemorrhage and massive fluid installation [18,19]. Selenium deficiency has been known to negatively affect immune cells during activation, differentiation, and proliferation [12]. Furthermore, low serum Se levels, with increased oxidative stress and inflammatory biomarkers, are frequently found in critically ill patients and are associated with poor prognosis [20]. Therefore, trauma patients with low serum Se levels may develop more infectious complications and, thus, bear a higher risk of adverse outcomes.

Nutrition support guidelines are available and provide suggestions for intensive care unit (ICU) settings [21]. These guidelines are adopted for patients with major trauma. However, the critical care population is heterogeneous, and guidelines might not apply to trauma patients. Patients who have sustained major trauma might have a unique demand for nutritional support; therefore, evidence-based recommendations for nutritional therapy in major trauma should be more specific. Recent nutritional therapy recommendations have suggested micronutrient substitution, including Se for patients with major burns [19]. To our knowledge, there are no critical and thorough analyses to support the efficacy of Se supplementation in severe trauma patients.

The objective of this review is to assess current evidence regarding the effectiveness of Se supplementation on mortality, length of ICU/hospital stay, and infection rates in patients having sustained trauma.

## 2. Materials and Methods

### 2.1. Protocol and Registration

A systematic literature search was performed based on the 2019 Preferred Reporting Items for Systematic Reviews and Meta-analyses (PRISMA) statement. This study was registered in Prospero (registered no. 297041).

### 2.2. Eligibility Criteria

Population.

This review considered studies that included acute trauma patients who had been admitted to hospital and received Se supplementation during hospitalization.

Intervention and Comparison.

Studies that evaluated nutrition support with Se, compared with placebo or standard treatment, were enrolled.

Outcome Measures.

Primary outcome measures for this review were mortality, length of stay (LOS; ICU/hospital), and complications (e.g., nosocomial infection, hospital-acquired pneumonia).

Studies.

This review considered randomized controlled trials (RCTs), experimental and epidemiological study designs (including non-randomized controlled trials, before and after studies), and prospective and retrospective cohort studies.

### 2.3. Information Sources and Search

We searched the MEDLINE, Embase, and Web of Science electronic databases. We confined the search to the English language and selected publication dates between January 1990 and December 2020. The following subject headings: “trace element”, “selenium”, “copper”, “zinc”, “injury”, and “trauma” were used. Table 1 provides these details. Moreover, the references of relevant articles were evaluated for other eligible studies.

### 2.4. Study Selection

The titles and abstracts were screened by two independent reviewers (J. F. Huang and C. P. Hsu) to determine the suitability of the studies for inclusion. Only studies that could provide data were used, since this allowed for calculating the odds ratio and relative risk factors. Furthermore, case reports and editorials were excluded.

### 2.5. Data Collection Process and Quality Assessment

Two reviewers (J. F. Huang and C. P. Hsu) independently extracted the data. Data included details about the interventions (dosage, route, duration, and regimen), demographics (age, gender), study methods (study design, case numbers), and outcomes (ICU length of stay (LOS), hospital LOS, morbidity, and mortality). When encountering missing data, these data were calculated using information from the publications.

The authorship or institution was not blinded to the reviewers. Disagreements regarding the entire review process were consulted by a third reviewer (C. H. Liao). The modified Jadad scale was used to assess the quality of the studies.

### 2.6. Data Synthesis

Quantitative data were pooled for the meta-analysis with the Comprehensive Meta-Analysis (CMA) 3.0 software (Biostat, Englewood, NJ, USA). Effect sizes were reported as odds ratios (OR) with 95% confidence intervals (CI) for mortality and infections. The standard difference in means (SMD) and their 95% CI for LOS were calculated. Heterogeneity was assessed statistically using the standard χ2 test and inconsistency quantified by the *I*^2^ test. Owing to the small number of included studies, publication bias was not tested, due to the insufficient power to detect real asymmetry [22]. A value of *p* < 0.05 was regarded as statistically significant.

## 3. Results

### 3.1. Study Selection

Through database searching, we identified 2671 citations. Twenty full-text articles were assessed for eligibility, 13 of which were rationally excluded. The remaining seven studies met the methodological quality criteria and were included in the meta-analysis [10,23,24,25,26,27,28]. The study selection process is illustrated in Figure 1.

### 3.2. Study Characteristics

The included studies were four prospective, randomized, blinded, control trials [10,23,24,27] and three non-randomized experimental trials [25,26,28]. Characteristics of the included studies and the extracted data are disclosed in Table 2. The studies in this review had 4564 participants.

### 3.3. Analysis

Six studies reported the impact of Se supplementation on the mortality rate of patients sustaining major trauma [10,23,24,25,27,28]. The pooled results showed that Se supplementation was associated with a lower mortality rate (Figure 2, OR 0.733, 95% CI: 0.586, 0.918, *p* = 0.007; heterogeneity, *I*^2^ = 0%). Regarding the incidence of infectious complications, there was no statistically significant benefit after analyzing the four studies (Figure 3, OR 0.942, 95% CI: 0.695, 1.277, *p* = 0.702; heterogeneity, *I*^2^ = 14.343%) [23,24,26,27].

Six studies disclosed intensive care unit (ICU) and hospital stays [10,23,24,25,27,28]. The patients with Se supplementation had shorter ICU lengths of stay (Figure 4, standard difference in means (SMD): −0.324, 95% CI: −0.382, −0.265, *p* < 0.001; heterogeneity, *I*^2^ = 0%). The hospital length of stay was also shorter with Se supplementation. (Figure 5, SMD: −0.243, 95% CI: −0.474, −0.012, *p* < 0.001; heterogeneity, *I*^2^ = 45.496%).

## 4. Discussion

In this meta-analysis, Se supplementation for severe trauma patients was examined. The current evidence supports that Se administration decreases the mortality rate and ICU and hospital stay for patients who have sustained major trauma. The pooled analysis suggested that Se supplementation was not associated with infectious complications after major trauma.

Deceased mortality was observed in patients administered Se supplementation (OR 0.733, 95% CI: 0.586, 0.918, *p* = 0.007). As the trimodal of mortality has shown [1], the mortality of trauma patients is initially related to injury severity. The protective effect of Se might reduce the inflammatory responses after severe trauma and, accordingly, reduce the mortality in the recovery phase. There were only limited numbers of studies with relatively small sample sizes, so the effect on mortality should be interpreted with caution. Moreover, the large cohort could influence the result for the study group. The hypermetabolic response, trace element losses following hemorrhage, and fluid resuscitation are unique to trauma. Some studies have reported that supplementary Se did not improve survival in critically ill patients [29,30]. In these studies, Se supplementation was provided to random critically ill patients without considering the patients’ conditions. In contrast, some authors reported that Se administration could be beneficial for patients with Se deficiency whether in severe burn or multiple trauma [15,31].

In this review, we identified the statistically significant effect of Se on length of hospital or ICU stay. It is reasonable to conclude that Se can reduce the inflammatory process during acute injury and shorten the recovery phase. The cost savings for the decrease in LOS outweigh the cost of the intervention. Further cost and benefit analyses would be worthwhile to assist clinicians and patients to understand the value of nutritional therapy. However, in the current review, the immunomodulation effect of Se did not translate to a reduction in infections. Therefore, the underlying mechanism should be clarified.

There were two studies that provided serum Se levels and addressed the effective dose range. However, the actual effective dose remains uncertain. Brätter et al. reported an estimated safe dose of 500 mcg/day for Se supplementation to avoid inhibition of the hepatic deiodinase. [32] Thus, the two studies used 500 mcg/day and 540 mcg/day as a supplement and found a significant increase in the serum Se level after a five-day intervention. Furthermore, the Se supplementation was associated with incremental thyroid hormone and glutathione peroxidase concentrations. Although not considered direct evidence, based on the available evidence, a daily Se dose of 500 mcg for five days might be suitable for patients who have sustained major trauma. Furthermore, all included studies in this review used a mixture of nutritional supplements rather than Se alone. In addition, the laboratory results also offered a clue regarding the effect of Se in the experimental mixture.

Recent investigations reported that a higher serum Se level was associated with hyperglycemia and type 2 diabetes mellitus (T2DM) [33,34,35,36]. Moreover, there were some conflicting studies regarding dietary Se supplementation and the risk of T2DM. Several cohorts disclosed that dietary Se supplements carried a risk of T2DM [35,37]. Nevertheless, some analyses showed a neutral or positive effect of Se supplementation on T2DM [38,39]. Coincidentally, higher serum Se levels were found to be related to dyslipidemia [40,41]. More efforts should be made to clarify the role of Se in metabolic diseases. Moreover, clinicians should bear in mind the metabolic effect of Se and ensure that patients supplemented with Se receive proper monitoring of their metabolic parameters during nutrition therapy.

Limited studies reported the use of Se supplementation in major trauma patients and lacked large-scale clinical trials. Importantly, this review contributed by providing evidence to this field. However, there were limitations in this review. First, when interpreting this review, the small number of studies with small sample sizes introduced possible type II errors. Second, since the included studies spanned 20 years, changes in trauma management over time could be another confounding factor. Third, variations in dosages and the mixture of antioxidants mask the actual effects of Se supplementation. Moreover, only two studies provided serum test results, and it was difficult to ensure the clinical benefit of Se supplementation and the underlying mechanism. In the future, large multicenter studies that consider injury severity, Se dosage, metabolic effects, and serum markers are essential to provide definitive evidence.

## 5. Conclusions

This review indicates that Se supplementation after severe trauma confers positive effects in decreasing the overall lengths of ICU and hospital stay. This review also indicates that Se supplementation appears safe and does not increase mortality or other reported adverse side effects.

## Figures and Tables

**Figure 1 nutrients-14-00342-f001:**
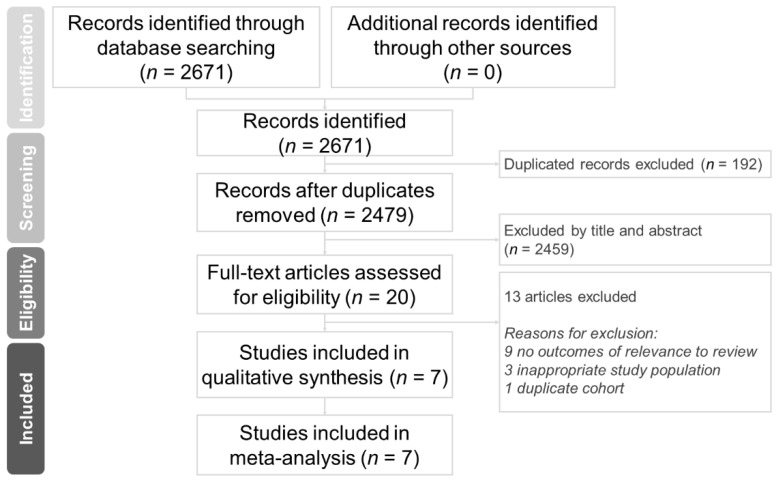
Protocol of this systematic review.

**Figure 2 nutrients-14-00342-f002:**
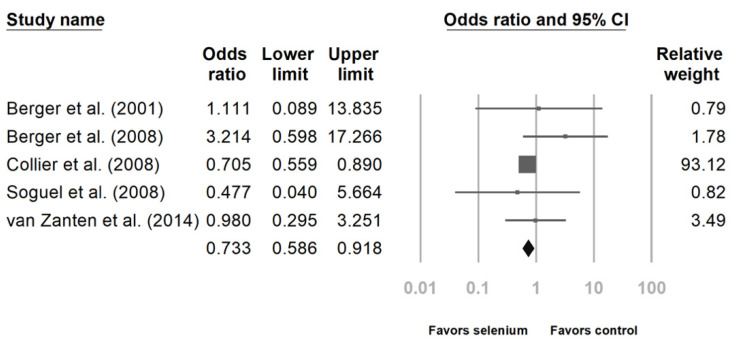
Forest chart of selenium (Se) supplementation on mortality of severe trauma patients.

**Figure 3 nutrients-14-00342-f003:**
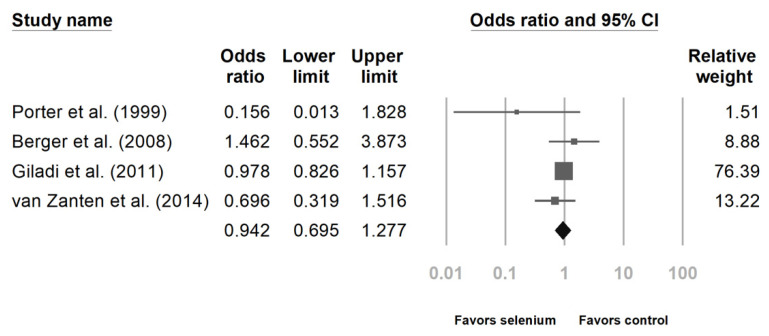
Forest chart of selenium (Se) supplementation on infection occurrence in severe trauma patients.

**Figure 4 nutrients-14-00342-f004:**
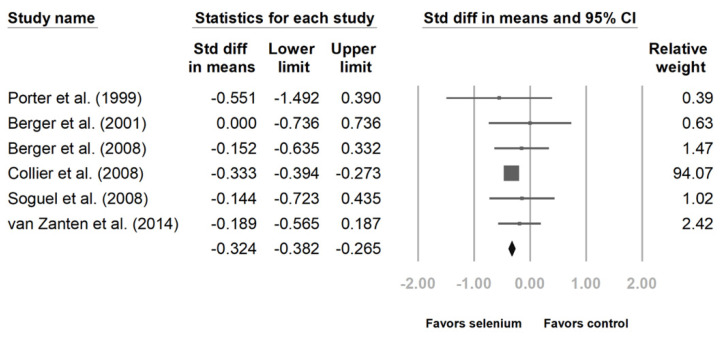
Forest chart of selenium (Se) supplementation on the length of intensive care unit stay of severe trauma patients.

**Figure 5 nutrients-14-00342-f005:**
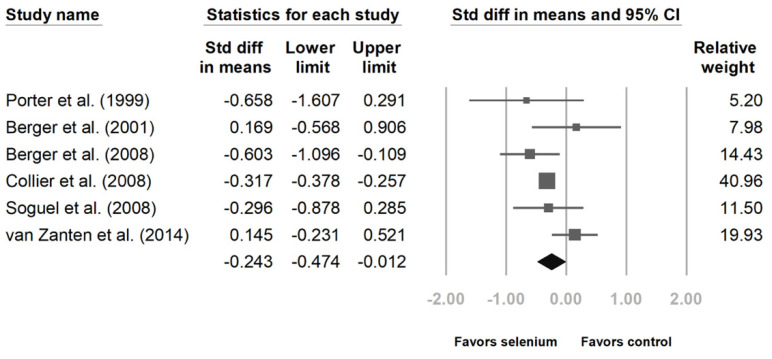
Forest chart of selenium (Se) supplementation on the length of hospital stay of severe trauma patients.

**Table 1 nutrients-14-00342-t001:** Search strategy including keywords and filters/limits.

Database	Search Terms	Filters/Limits
PubMed	(trace element OR selenium OR copper OR zinc) AND (trauma OR injury)	Clinical Trial, Randomized Controlled Trial, Humans, English, 1990–2021
Embase	(trace:ti,ab AND element:ti,ab OR selenium:ti,ab OR zinc:ti,ab OR copper:ti,ab) AND (trauma:ti,ab OR injury:ti,ab)	(humans)/lim AND (English)/lim AND (clinical study)/lim AND (1990–2021)/py
Web of Science	((TS = (trace element OR selenium OR copper OR zinc)) OR (TI = (trace element OR selenium OR copper OR zinc))) AND ((TS = (trauma OR injury)) OR TI = (trauma OR injury)) NOT ALL = (in vitro OR rabbit OR rat OR animal OR mice OR mouse OR pig OR porcine OR sheep OR lamb) AND (DT = (“ARTICLE” OR “MEETING ABSTRACT” OR “PROCEEDINGS PAPER” OR “EDITORIAL MATERIAL” OR “EARLY ACCESS”))	1990–2021

**Table 2 nutrients-14-00342-t002:** Characteristics of included studies.

Study	Methods	Participants, Setting	Intervention	Outcome Measures	Modified Jadad Scale
Porter et al. (1999)	Study design: prospective RCT Duration of follow-up: until discharge	Participants: Total *n* = 18 Intervention *n* = 9; placebo *n* = 9 18 males Mean age: 32.4 years Setting: ICU, Lincoln Medical Center, Bronx, New York Inclusion criteria: age 15–80; penetrating injury involving multiple systems, ISS not <25 or abdominal trauma index not <25, or any two of the following: prehospital and admission SBP <90, initial right atrial venous oxygen tension of <35 mm Hg and O_2_ saturation of <65%, base excess worse than −8, or serum lactate >4	Intervention: 50 mg of selenium IV, Q6H, 400 IU of vitamin E PO Q8H, 100 mg of vitamin C PO Q8H, and 8 g of *N*-acetylcysteine (NAC) PO Q6H for 7 days, stay in ICU or until death, whichever one was shorter Control: Standard therapeutic care	Mortality, incidence of septic complications, ARDS, MODS, hospital stays	6.5/8
Berger et al. (2000)	Study design: prospective RCT, DB Duration of follow-up: until discharge	Participants: Total *n* = 31 Intervention *n* = 20; placebo *n* = 11 23 males and 8 females Mean age: 42 ± 16 years Setting: ICU, Centre Hospitalier Universitaire Vaudois, Lausanne, Switzerland Inclusion criteria: Age 18–75; multiple injuries with ISS >15; admission within first 24 h of injury	Intervention: Selenium 500μg IV QD, with or without vitamin E 150 mg IV QD, Zn 13 mg IV QD for 5 days Control: placebo	Mortality, incidence of complications and organ failure, hospital stays	8/8
Berger et al. (2008)	Study design: prospective RCT, DB Duration of follow-up: 3 months after discharge	Participants: Total *n* = 66 Intervention *n* = 34; placebo *n* = 32 52 males and 14 females Mean age: 40 ± 19 years Setting: ICU, Centre Hospitalier Universitaire Vaudois, Lausanne, Switzerland Inclusion criteria: ISS >9	Intervention: selenium 270μg IV, zinc 30 mg IV, vitamin C 1.1 g IV, vitamin B_1_ 100 mg IV, vitamin E 6.4 mg IV and 300 mg PO with a double-loading dose on days 1 and 2, total for 5 days plus ICU standard vitamin profile (as for control group) Control: ICU standard vitamin profile: 500 mg vitamin C/day for 5 days and 100 mg vitamin B1/day for 3 days	Mortality, hospital stays, kidney function, subsequent organ failure, infections and pneumonia	8/8
Collier et al. (2008) Giladi et al. (2011)	Study design: Retrospective cohort study Duration of follow-up: Until discharge	Participants: Total *n* = 4294 Patients after AO protocol, *n* = 2272 Patients before AO protocol, *n* = 2022 3284 males and 1010 females Mean age: 40 years Setting: Level I Trauma Center, Vanderbilt University Medical Center Inclusion criteria: All admitted trauma patients	AO protocol for all admitted trauma patients: selenium 200μg IV QD, vitamin C 1 g IV Q8H, vitamin E 1000IU PO Q8H for 7 days or until hospital discharge, whichever was shorter. Control: Patients admitted to the trauma center one year before implementation of AO protocol	Mortality, hospital stays, development of organ failure or dysfunction, infectious complications	2/8
Soguel et al. (2008)	Study design: Prospective study with historical control Duration of follow-up: Until discharge	Participants: Total *n* = 40 Intervention *n* = 20; control *n* = 20 30 males and 10 females Mean age: 43.5 ± 19.2 years Setting: ICU, Centre Hospitalier Universitaire Vaudois, Lausanne, Switzerland Inclusion criteria: Trauma patients needed ICU care	Intervention: selenium 300μg PO QD, zinc 30 mg PO QD, vitamin C 1.5 g PO QD, vitamin E 500 mg PO QD for 10 days; vitamin B_1_ 100 mg IV QD, vitamin C 500 mg IV QD for 5 days Control: historical matched controls matching criteria: age, sex, ISS, brain injury, and SAPS II score.	Sequential organ failure assessment (SOFA) score, mortality, hospital stays, infection complications	4/8
van Zanten et al. (2014)	Study design: prospective RCT, DB Duration of follow-up: 6 months after start of study product	Participants: Total *n* = 109 Intervention *n* = 55; control *n* = 54 87 males and 22 females Mean age: 43 years Setting: ICU, multi-country, multi-center Inclusion criteria: age ≥18 years, mechanically ventilated ICU patients	Intervention: Tube feed formula enriched in glutamine, vitamins C and E, selenium, zinc, and EPA and DHA and low in carbohydrate content maximum for 28 days during ICU stay Control product: Isocaloric standard tube feed with the same amount of protein.	Incidence of nosocomial infections and organ failure, duration of ventilation, ICU, and hospital stay, mortality	8/8

RCT: randomized control trial; ICU: intensive care unit; ISS: injury severity score; SBP: systolic blood pressure; DB: double blinded; AO: antioxidant; EPA: eicosapentaenoic acid; DHA: docosahexaenoic acid.

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
