# Peer review of "The Impact of Selenium Supplementation on Trauma Patients—Systematic Review and Meta-Analysis"

_nutrients, 2022, doi:10.3390/nu14020342_

Round 1
Reviewer 1 Report
- This study resulting evidence supports parenteral supplementation of Se administration in decreasing mortality rate, ICU and hospital stay in severe trauma patients. The pooled analysis of individual study data suggests that combined Se supplementation did not increase the morbidity, and the presence of hospitalized acquired pneumonia after severe trauma.. It is very interesting. Moreover, it would be better to summarize the effective dose range and addition time of selenium for severe trauma in discussion.
- Try to provide the serum selenium content before and after intervention?
- There are some cohort studies on the intervention of selenium in nutritional metabolic diseases. Make a comparative study in the discussion.
- “Se” and “Selenium” should be uniformed in the whole text. Such as line 194, 195, 197, and so on.
- The format of references is not uniform. At the same time, more new literatures need to be cited.
Author Response
This study resulting evidence supports parenteral supplementation of Se administration in decreasing mortality rate, ICU and hospital stay in severe trauma patients.
The pooled analysis of individual study data suggests that combined Se supplementation did not increase the morbidity, and the presence of hospitalized acquired pneumonia after severe trauma. It is very interesting.
Comment 1: Moreover, it would be better to summarize the effective dose range and addition time of selenium for severe trauma in discussion.
Reply: Thank you for your thoughtful suggestion. There were two studies providing serum selenium levels and addressing the effective dose range. However, the actual effective dose remained uncertain. Brätter et al. reported an estimated safe dose of 500mcg/day for selenium supplement to avoid inhibition of the hepatic deiodinase. Thus, the two studies had used 500mcg/day and 540mcg/day as a supplement and found a significant increase in serum selenium level after the 5-day intervention. Furthermore, the supplementation was associated with incremental thyroid hormone and glutathione peroxidase concentrations. Though not direct evidence, daily 500mcg selenium for five days might be suitable for the patients sustaining major trauma based on the current evidence.
Comment 2: Try to provide the serum selenium content before and after intervention?.
Reply: Thank you for your critical comments. There were two studies providing serum selenium levels and addressing the effective dose range. The two studies had used 500mcg/day and 540mcg/day as a supplement and found a significant increase of serum selenium level after the 5-day intervention. Furthermore, the supplementation was associated with incremental thyroid hormone and glutathione peroxidase concentrations. We have added this information to the revised manuscript.
Comment 3: There are some cohort studies on the intervention of selenium in nutritional metabolic diseases. Make a comparative study in the discussion.
Reply: Thank you for the valuable comment. Recent investigations reported higher serum Se level was associated with hyperglycemia and type 2 diabetes mellitus (T2DM). There were some conflicting studies regarding the dietary Se supplement and the risk of T2DM. Coincidentally, higher serum Se level was found to be related to dyslipidemia. More efforts should be made to clarify the role of Se on metabolic diseases. Moreover, the clinician should bear the metabolic effect of Se in mind and ensure the patients under Se supplement receive proper monitoring of the metabolic parameters during the nutrition therapy. We have put the above observations in the discussion. Thank you for the advice to make the article complete.
Comment 3: “Se” and “Selenium” should be uniform in the whole text. Such as lines 194, 195, 197, and so on.
Reply: Thank you for your comments, we unified the phrase in the whole text to make it clear to read.
Comment 4: The format of references is not uniform. At the same time, more new literature needs to be cited.
Reply: Thank you for your comments, we corrected the format of references. Thank you again for your advice to make our manuscripts worth reading.

Reviewer 2 Report
I recommend to adjust the manuscript because in most of the selected publications for the meta-analysis a combination of selenium and other nutrients like copper, zinc, vitamin B1, E and C and/or EPA/DHA were administered. To seperate the role of selenium next to the role of the other nutrients is not well explained. A draw back of the meta-analysis is also that the nutrient status, especially selenium status, at the start of the intervention is not reported. So the question to be discussed is whether the benefit is only in those who have a low selenium status or the benefit is ude to the high dose supplementation.
Author Response
Comment 1: Recommend to adjust the manuscript because in most of the selected publications for the meta-analysis a combination of selenium and other nutrients like copper, zinc, vitamin B1, E and C and/or EPA/DHA were administered. To separate the role of selenium next to the role of the other nutrients is not well explained.
Reply: Thank you for your constructive comments. Like your words, most of the included studies were combined usage of the trace nutrients rather than Se itself. Actually, there was no available study that presented an isolated Se supplement in trauma patients. Concerning the complicated antioxidant mechanism, no single element could play the role well without other nutrients. We assumed this might be the reason why most studies used a mixture supplement rather than Se alone. There were two studies describing a significant increase of serum selenium level after the 5-day intervention. The supplementation was associated with incremental thyroid hormone and glutathione peroxidase concentrations. This could be indirect evidence about the role of Se in the experimental mixture. We have added this information in the revised manuscript to make it clear to read. Further studies would be needed to clarify the importance of each trace element.
Comment 2: A drawback of the meta-analysis is also that the nutrient status, especially selenium status, at the start of the intervention is not reported. So the question to be discussed is whether the benefit is only in those who have a low selenium status or the benefit is due to the high dose supplementation.
Reply: Thank you for your critical comments. As stated before, there were two studies describing a significant increase of serum selenium level after the 5-day intervention. The supplementation was associated with incremental thyroid hormone and glutathione peroxidase concentrations. We have added this information to the revised manuscript. Thank you again to make our manuscript value to read.

Round 2
Reviewer 2 Report
I am satisfied with the feedback.